



# Accuracy assessment of real-time flood forecasting of coupled hydrological and mesoscale meteorological models

Aida Jabbari[1], Jae-Min So[1], Deg-Hyo Bae[1]

[1]Department of Civil and Environmental Engineering, Sejong University, Seoul 05006, Korea

*Correspondence to*: Deg-Hyo Bae (dhbae@sejong.ac.kr)

**Abstract.** Hydro-meteorological predictions are important for water management plans, which include providing early flood warnings and preventing flood damages. This study evaluates the real-time precipitation of an atmospheric model at the point and catchment scales to select the proper hydrological model to couple with the atmospheric model. Furthermore, a variety of tests were conducted to quantify the accuracy assessments of coupled models to provide details on the maximum

spatial and temporal resolutions and lead times in a real-time forecasting system. As a major limitation of previous studies, the temporal and spatial resolutions of the hydrological model are smaller than those of the meteorological model. Here, through ultra-fine scale of temporal (10 min) and spatial resolution (1 km × 1 km), we determined the optimal resolution. A numerical weather prediction model and a rainfall runoff model were employed to evaluate real-time flood forecasting for the Imjin River (South and North Korea). The comparison of the forecasted precipitation and the observed precipitation

indicated that the Weather Research and Forecasting (WRF) model underestimated precipitation. The skill of the model was relatively higher for the catchment than for the point scale, as illustrated by the lower RMSE value, which is important for a semi-distributed hydrological model. The variations in temporal and spatial resolutions illustrated a decrease in accuracy; additionally, the optimal spatial resolution obtained at 8 km and the temporal resolution did not affect the inherent inaccuracy of the results. Lead time variation demonstrated that lead time dependency was almost negligible below 36 h.

With reference to our case study, comparisons of model performance provided quantitative knowledge for understanding the credibility and restrictions of hydro-meteorological models.

## 1 Introduction

One of the most expensive natural disasters is due to severe floods, which that are often triggered by heavy precipitation.

Hydro-meteorological predictions are important for providing early flood warnings and preventing or reducing flood damages. Accordingly, using hydrological models for rainfall-runoff estimation has become increasingly practical and popular; however, improvements in the capabilities of Numerical Weather Prediction (NWP) models increased the tendency to use forecasting rainfall. Coupling the NWP and hydrological models connects the progress in meteorology and hydrology to generate real-time flood forecasting. In one-way linking of meteorological and hydrological models, the atmospheric



model outputs were extracted and used as input data for hydrological models. The Quantitative Precipitation Forecast (QPF) can be an alternative input data source for hydrological predictions.

Hydrological model developments range from lumped to semi-distributed and fully distributed models. Within this wide range, the selection of the proper model for different purposes can be a difficult task, which depends on user experience and feasibility (Todini, 2007). When choosing a hydrological model for flood forecasting, the accuracy of the precipitation which is vast from the pixel and subbasin to the catchment scale plays a major role in selection. Evaluation of point precipitation and the Mean Areal Precipitation (MAP) could lead to robust decision making in the distributed (which uses the point precipitation data as input) and semi-distributed (which uses the MAP as input) hydrological models. To clarify these aspects, the effect of precipitation errors on forecast flow is reported in previous studies (Moulin et al., 2009; Bárdossy and Das, 2008).

In the evaluation of meteorological models, it has been found that atmospheric models have difficulty to forecast accurately their space-time evolution therefore; atmospheric models predict the occurrence of the rainfall better than the magnitude and location of rainfall (Ebert and McBride, 2000; Cuo et al., 2011). Generally, NWP models overforecast light (1 mm) to moderate (5 mm) precipitation; however, they underforecast heavy precipitation in the Middle Atlantic Region (MAR) of the USA (Siddique et al., 2015) as well as in the wet and high elevation areas of the Ovens catchment in Australia (Shrestha et al., 2013). The evaluation of forecast precipitation values indicated that the accuracy of rainfall forecast varies with the spatial resolution (Chang et al., 2004; Koyabashi et al., 2016; Misenis and Zhang, 2010; Queen and Zhang, 2008; Weisman et al., 2008), temporal resolution (Ochoa-Rodriguez et al., 2015; Wetterhall et al., 2011) and lead time (Jang and Hong, 2014; Ghile and Schulze, 2010; Lin et al., 2010; Buizza et al., 1999) of meteorological models. The accuracy of atmospheric variables improved by increasing the resolution of the Fifth-Generation Pennsylvania State University National Center for Atmospheric Research Mesoscale Model (MM5) using 36, 12, and 4 km spatial resolutions for the Snoqualmie River in Washington (Westrick and Mass, 2001). Moreover, the effect of forecasting lead time on the accuracy of predicted values showed that the accuracy of the predictions and the forecast capabilities significantly improved by decreasing the forecasting lead time (Chang et al., 2004). The comparison of different lead times for WRF model forecast showed that increasing the lead time caused the overestimation of rainfall in the Liujiang River basin and decreased the forecast accuracy (Li et al., 2017).

Previous studies simulated or forecasted stream flow using WRF model data that were forced to the hydrological models (Givati et al., 2012; Kumar et al., 2008; Shih et al., 2014). Coupling WRF with Hydrological Model for Karst Environment (HYMKE) in the Jordan River basin showed good agreement between forecasted stream flow and measured stream flow (Givati et al., 2012). A comparison of the one-way and two-way coupling of the WRF and Hydrological Engineering Center-Hydrological Modeling System (HEC-HMS) model indicated the case had improved forecasting ability (Givati et al., 2016). In real-time hydro-meteorological studies, one limitation of previous studies is that the time/spatial scale of the hydrological model is much finer than that of the meteorological model. Furthermore, there are few studies in the literature about the quantification of real-time forecast precipitation analysis.



In this study, an ultra-fine temporal and spatial resolution scale provided an opportunity to more deeply analyze and improve our understanding of the effect of lead time, spatial and temporal resolution variation on the performance of coupled hydro-meteorological models. The aim of this study is to evaluate the real-time precipitation of the atmospheric model at the point and catchment scales in order to select a hydrological model for coupling with an atmospheric model. Moreover, the

accuracy assessment of a coupled hydro-meteorological model is done for a real-time system to find how the variations in spatial and temporal resolution and lead time are reflected in the precipitation and flood forecasting. To achieve this goal, a variety of tests were conducted to quantify the accuracy of discharge and precipitation. This research provides details on the maximum spatial and temporal resolutions and the lead time required for reliable forecasts in a real-time forecast system.

## 2 Study area and data

Documents and analysis will entirely focus on the Imjin River, the seventh largest river in Korea, which passes through North and South Korea. The area and the length of this domain are 8139 km$^2$ and 273.5 km, respectively. The river originates in North Korea, heads from the Hamgyeongnam-do Masikryoung Duryu mountain and flows from North to South passing the demilitarized zone (DMZ), and joins the Han River and finally the Yellow Sea. The average annual precipitation is approximately 1100 mm, and the topography varies from 155 m to 1570 m above mean sea level. Since two-thirds of the

Imjin River is located in North Korea, this river is considered a transboundary river. Given that immediate access to data in Transboundary Rivers is hard due to political boundaries and data reliability, Transboundary Rivers are always challenging for engineers and model developers. Therefore, it was difficult to obtain the required information for the hydrological model in the northern part of the basin. In addition, the study area includes 38 subbasins. The subbasins and location of water level gauges are shown in Figure 1, which provides a broad depiction of the study area. Other applications of the flood forecasting

system are greatly related to military operations. Similarly, the Imjin River basin has a military region named Paju-si. This region includes river-crossing operations. Flood hazards directly influence military operations and require the use of detailed and widely applicable hydrological models. The coupling of hydrological and meteorological models is done to enhance flood information analysis in this important area.

Heavy rainfall and repeated storms have been reported many times in South and North Korea, especially during the summer

and fall (Lee et al., 2013). The Imjin River has encountered various flood events across several years. The extreme events are chosen for consideration in flood forecasting in the Imjin River basin. The 2002 flood event is represented by 20020828-0904, the 2007 flood event is represented by 20070723-0904, and the 2011 flood event is represented by 20110725-0730. In 2002, Typhoon Rusa ripped through South Korea in the Gangneung area between 31 August and 1 September, affecting the eastern and southern parts of Korea with almost 900 mm of rainfall in 30 hours (Kim et al., 2006). The typhoon caused the

submergence of 9000 houses and killed 113 people. In 2007, North Korea had heavy rain between August 7-14 and September 18-20 (caused by Typhoon Wipha). Over seven days, approximately 500-700 mm of rainfall caused this flood in North Korea. Seoul also experienced heavy flooding on July 27, 2011. In the case of fast growing densely populated cities



such as Seoul, flash floods of 536 mm of rainfall in three days resulted in 69 people reported as dead or missing in the Gangnam area. These intense rainfall events led to hazardous floods and caused various damages in South and North Korea. The floods were caused by torrential rainfall in this area and indicate the need for integrated flood management, especially for two countries with different natural environments, national defenses and political problems. In this study, the hydro-

5    meteorological components are coupled for real-time rainfall-runoff forecasting procedures for the transboundary Imjin River. There are 66 rain gauges for the events 2007 and 2011 and 33 rain gauges for the event 2002, and there are three meteorological stations in the Imjin basin. The observation data used in this study were passed through a quality control procedure, which checked the values and filled missing values by interpolating from nearby stations to complete the hourly data from all stations.

## 3 Methodology

### 3.1 Meteorological model

The WRF model is a mesoscale NWP model designed for atmospheric and operational forecasting research. The WRF includes two dynamical cores, data assimilation, and software. The WRF model applies to a wide range of meteorological issues at various scales. The application of the WRF model as a mesoscale model for forecasting extreme events has been

used worldwide. This model enables researchers to conduct different atmospheric simulations, such as real data or idealized conditions, by providing operational forecasting, including advances in physics, numeric, and data assimilation. The initial and boundary conditions were obtained using external sources, such as the static geographic data provided by the USGS and MODIS data set and the gridded data provided by regional and global models such as the North American Mesoscale Forecast system (NAM) and the Global Forecast System (GFS) (Skamarock et al., 2008). The weather charts were provided

by the National Centers for Environmental Prediction (NCEP) Final Analysis (FNL), which has a resolution of $1° \times 1°$.

In summary, using the definitions of all computational grids, geogrid interpolates terrestrial, time invariant fields, and then Ungrib extracts the meteorological fields from the GRIB formatted files, and Metgrib horizontally interpolates the meteorological data to the simulation domains. The Advanced Research WRF (ARW) solver uses time-splitting techniques to integrate the fully compressible non-hydrostatic equations of motion. The Euler equations are in flux form and are

formulated using a terrain that follows mass vertical coordinates. Finally, time-split integration is carried out using the second or third order Runge-Kutta method (Skamarock et al., 2008). In this study, the WRF version 3.5.1 was used for real-time forecasting of the meteorological data by using the WRF Double-Moment 6-Class (WDM6) microphysical scheme. On the meteorological side, the WRF model covered Korea and the surrounding region with a high temporal and spatial resolution. The model provided forecasts of 10 min data over a 72 h time series repeated every 6 h for the following real-

time meteorological parameter fields: precipitation, temperature, wind speed, relative humidity, and solar radiation. The WRF model is configured on a Mercator projection with $400 \times 400 \times 40$ grid points and an ultra-fine scale resolution of 1 km × 1 km.





### 3.2 Hydrological model

To find the proper hydrological model, the assessment of the forecasted precipitation for rain gauge stations and MAP are done using individual forecasts and the mean of the forecast data (Fig. 2). The quality of forecast precipitation can be analyzed by comparing the values with observation data. For this purpose, precipitation analysis can be done by the average ensemble method using equal weighting to the members, which are lagged by 6 hours. This approach included forecast averages from multiple lead times, which were then compared with observed data. The methodology used to compare the observation and with the average of the real-time ensemble forecast data is depicted in Figure 3. The Root Mean Square Error (RMSE) is used for the comparison between real-time forecast data and the observed data. The RMSE is one of the most widely used approaches for verification, and it evaluates the average magnitude of real-time forecast errors. The outcomes from this analysis are provided in the Results section (please refer to Section 4.2), and the findings resulted in the selection of the semi-distributed hydrological model that is described as follows.

The Sejong University Rainfall Runoff model (SURR) was developed by the Water Resource and GIS Laboratory, Sejong University (Bae and Lee, 2011). The model was developed based on the event-oriented storage function model (Kimura et al., 1961). The SURR model is a semi-distributed continuous rainfall runoff model that improved the estimation of hydrological components such as potential evapotranspiration, surface flow, lateral flow, and groundwater flow using physical foundations. This model was developed to illustrate the complicated and nonlinear relationship between rainfall and runoff in combination with natural components, such as soil moisture condition and land use. The SURR model requires the input data as MAP and Mean Areal Evapotranspiration (MAE) for each of the sub-catchments. The SURR model can be driven by either observed or forecasted precipitation data. The FAO Penman-Monteith method was chosen as the method by which evapotranspiration was estimated from meteorological data. The rainfall and evapotranspiration data have hourly temporal resolutions, which were spatially interpolated by the Thiessen polygons method using GIS.

Hydrological model parameters and formulation affect the ability of the hydrological model to simulate the streamflow. Therefore, as an initial assessment, the calibration and verification of the hydrological model could be done using historical data to determine the stability of the model. In the SURR model, there are two types of incorporated parameters, which include the subjective and objective parameters. These parameters can be estimated based on the basin characteristics and are computed in the model calibration process. For rainfall runoff simulation, the sensitive parameters of the SURR model are indicated in Table 1. The SURR model was calibrated for the Imjin basin using the observed rainfall and streamflow, and the optimized parameters resulted in good agreement between the observed and simulated streamflow during the verification periods. The calibration periods were considered as the 2007 (20070701-0910) and 2008 (20080701-0910) events, while the verification periods were considered as 2009 (20090701-0910), 2010 (20100701-0910), 2011 (20110701-0910) and 2012 (20120701-0910) events. For the sake of the brevity, the results of the calibration and verification are shown for the 2008 and 2012 events for Jeonkuk station (Fig. 4). A detailed description of the SURR model is reported in Bae and Lee (2011).





### 3.3 Accuracy assessment

In flood forecasting of coupled hydro-meteorological models, each model component has its own source of producing errors spread from the atmospheric conditions to rainfall forecasts and the rainfall to runoff predictions (Davolio et al., 2008). In this research, the results were obtained based on a one-way coupling methodology that connected the SURR and WRF models. To further investigate the accuracy of the coupled hydro-meteorological models, it is essential to use a methodology that quantifies the errors of the coupled system. As mentioned before, this study does not focus on the errors related to the hydrological model performance; therefore, the overall procedure consists of quantifying the accuracy of the precipitation analysis, the spatial and temporal resolution and the variation in lead time using statistical measures.

The precipitation is the most important output of meteorological models used for hydro-meteorological applications since the performance of coupled hydro-meteorological models is dependent on the accuracy of forecasted precipitation. Therefore, it is necessary to establish a methodology to analyze the real-time forecast data. Point precipitation analysis provided a comparison between the observed precipitation at rain gauge stations and the values that were forecast for each place. Furthermore, the areal average values of precipitation were calculated by Thiesen polygons at the catchment scale using the observed precipitation at the rain gauge stations versus the high resolution of the meteorological model forecast data. The accuracy assessment of variations in spatial resolution (1, 2, 4, 8, 12, 16 and 20 km), temporal resolution (10, 20, 30 and 60 min) and lead time (12, 24, 36, 48, 60 and 72 h) can be performed by evaluating the precipitation and analyzing the discharge. The correlation, bias and RMSE can be used to show the level of agreement between the observed and forecasted values and the accuracy variation for the abovementioned items, respectively.

According to the results of this analysis, the optimal spatial and temporal resolution and lead time can be chosen to be used in the time series analysis of forecast flood. Evaluation techniques have been reviewed using a variety of gauge and forecasted data in the SURR and coupled SURR-WRF models. With these different types of comparisons, it is possible to establish the quality of each component. Comparisons of simulated, observed and forecast stream flow can be done using statistical indexes to quantify the accuracy assessments. The efficiency criteria used in this study are presented and evaluated (Table 2). These criteria include the Nash-Sutcliffe Efficiency (NSE) by Nash and Sutcliffe (1970), Mean Relative Error (MRE), and Relative Error in Volume (REV). These indexes provide more information on the systematic and dynamic errors present in the model results.

### 4 Results and analysis

### 4.1 Point precipitation assessment

In coupled hydro-meteorological models, for real-time flood forecasting, the error related to the rainfall forecasts overcomes the other sources of error (Diomede et al., 2008). In this part, the attention is focused solely on the meteorological model. Observed precipitation has a complicated nature, which makes it difficult to use for atmospheric validation, but it could be a



useful tool for detecting precipitation errors, such as those caused by the position, timing and strength of the events (Benoit et al., 2003). The amount of precipitation produced by the atmospheric model is compared with rain gauges to diagnose the forecast errors. Due to the high resolution of the WRF model in this study, the error related to locating different stations in the same model grid cell with different observations is eliminated. This analysis is applied for spot measured rainfall in

observation stations and for real-time forecasted data from the WRF model at the mentioned locations.

In this method, each observation is compared with a single corresponding mean forecast data from the same time. Precipitation analysis at the point scale shows that the skill of the NWP precipitation forecasts varies considerably between rain gauge stations. The results of the point precipitation assessment at the 33 stations for the 2002 event and the 66 stations for the 2007 and 2011 events indicated that the WRF model does not forecast the rainfall well. Further investigation is

conducted to compare the total, minimum, maximum and underestimation of observed and forecast rainfall for the duration of the events (Table 3). The scatter plot of observed and forecast precipitation showed that the WRF model less accurately captures the precipitation accurately in all events. In the present study, the WRF as a NWP model has the limitation of underestimating the precipitation in this study area (Fig. 5).

## 4.2 Spatial distribution of MAP

To have a general view of the meteorological models' performance, it is of great necessity to evaluate the NWP model performance at the catchment scale. The figures demonstrate the spatial patterns of MAP, which were obtained by the observed and forecasted precipitation for each subbasin in the Imjin watershed. The observation-based MAPs were calculated using observed precipitation, while the forecast MAPs were obtained from the WRF model. The differences can be detected in the intensities of the observed data and the WRF model data in the whole area. In the flood event of 2002,

there is an increasing pattern in the observed data in most of the southern region; however, the WRF model results indicated a decreasing N-S gradient and an increasing pattern in the center region. The results related to the 2007 event showed a positive northbound and southbound gradient of areal precipitation in the observed and forecasted data, respectively. In the 2011 event, the observed and WRF data had an increasing pattern of rainfall from North to South while the WRF results included an underestimation for the MAP. Taken together, at the catchment scale, the WRF model predicted the general

rainfall pattern well; however, it had some significant underestimation with respect to the observations (Fig. 6). The results of the MAP assessment showed that the WRF model had underestimated the MAPs by 84.2, 78.9 and 97.4% for the 2002, 2007 and 2011 events, respectively (Table 4).

The results of the MAP analysis for the 38 subbasins exhibited that the WRF model had better performance for the 2002 event, with a lower RMSE of 122 than the higher RMSE of 159 and 304 for the 2007 and 2011 events, respectively (Table 5).

Overall, the spatial averaging of rainfall over the catchment reduces the errors compared to point analysis. The accuracy assessment for the comparison of forecast precipitation and MAP indicated that significant differences can be expected for the distributed and semi-distributed hydrological models. According to the findings, the semi-distributed hydrological model may be a better choice for this study. For the SURR model, as a semi-distributed hydrological model used in this study,





better forecast stream flow can be expected as a result of the lower RMSE in MAP than that from the point precipitation accuracy assessment.

### 4.3 Spatial resolution assessment

It is necessary to assess the spatial resolution effect on the accuracy of coupling the WRF model with the SURR model. Theoretically, higher resolution modeling with better mathematical characterization of physical processes is expected to lead to more accurate forecasts (Mohan and Sati, 2016; Gego et al., 2005). The WRF model feeds the whole domain of study with a dense spatial resolution. By comparing the spatial resolutions in this study, the results implied that the hypothesis stating higher spatial resolution data have better accuracy is demonstrated by the clear trend of the increasing error percentage with the decreasing spatial resolution in the WRF model. The presentation of the results starts with the evaluation of MAP correlation, the bias of observed and forecast data, and the RMSE of streamflow analysis.

The recommended minimum spatial resolution is a factor related to the complexity of the study area. The complex terrain and mountainous areas require higher resolutions. The MAP correlation and bias between observed and forecast data are illustrated in Figure 7. The bias evaluates the difference between the mean of the forecast and observation data, and the correlation illustrates the linear relation among the forecast and observation data. The results indicated that, by decreasing the spatial resolution, the bias increased, and the correlation coefficient decreased for all events. Further analysis led to evaluate the effect of the variation in spatial resolution on the real-time flood forecasting to choose the optimal resolution for coupling the WRF model with the SURR model. To examine the effects of spatial resolution variation on streamflow, the observed and forecasted flows are compared for different spatial resolutions. Increasing the spatial resolution of the meteorological models yielded improvements in the forecasted streamflow (Fig. 8). It is determined that the spatial resolutions lower than 8 km did not affect the inherent inaccuracy of the flood forecasts in all events, while after that, the error increased to a higher level for all events. It can be concluded that the WRF model is more likely to resolve physical procedures at higher spatial resolutions.

### 4.4 Temporal resolution assessment

The comparison of variations in temporal resolution of the WRF model forecast with observations for MAP correlation, bias and discharge analysis are indicated in Figures 9 and 10. In addition, as is seen, the accuracy assessment of temporal resolution showed that the performance did not change much by increasing the temporal resolution. The accuracy assessment is implemented to check the temporal resolution variation for 10, 20, 30 and 60 min data. The results of the bias evaluation for the MAP indicated that the bias did not change significantly for the different temporal resolutions. However, for the 2011 event, the bias is higher than the other events, and this could be related to the underestimation by the WRF model. The results of the MAP assessment in the previous sections (Table 4) showed underestimations of 97.4, 84.2 and 78.9% for the 2011, 2002 and 2007 events, respectively. The results of the MAP correlation assessment showed that the correlation did not vary significantly for the three events. The findings of the RMSE assessment for flood forecasting illustrated that the



temporal resolution variation did not affect the RMSE significantly. Generally, the results of the MAP correlation and bias, along with the error measurement of forecast discharge by RMSE did not vary for the different temporal resolutions in all events.

## 4.5 Lead time variation assessment

Lead time is a key factor in the NWP model forecasts since the skills of the models vary significantly with the forecast lead time. In general, the forecast skill decreases with the increase in lead time, which is related to the higher uncertainty in the forecast data. In fact, the NWP models underpin many statistical and hybrid techniques and typically use global-scale models to provide boundary conditions. Accordingly, the NWP models are influenced by the spin-up effect, which results in deficiencies during the first few hours of the forecast lead times.

The error measurement of each lead time (L) is given by Eq. (1):

$$RMSE_L = \sqrt{\frac{1}{N}\Sigma\left(Q_{t+L} - \hat{Q}_{t+L}\right)^2} \qquad\qquad L = 12,\ 24,\ 36,\ 48,\ 60,\ 72\ \text{h} \qquad\qquad (1)$$

where N is the number of discharges, $\hat{Q}_t$ is the forecasted discharge at time t obtained by the coupled SURR and WRF model, $Q_t$ is the observed streamflow, and L is the lead time. The summations are for all forecasts, which are the forecast time *t* belonging to all events. Further analysis compared the performance of the WRF model and the coupled SURR-WRF models

with different lead times. The results showed that the error measurements deviated with changes in lead time. The accuracy of the model results depended on the forecast lead time. This indicated that real-time forecasting systems performed better with short forecast lead times than with longer ones. The results of the observed and forecast rainfall comparison using a scatterplot indicated the over- and underestimation of the forecast rainfall for different lead time intervals (Fig. 11). The ensemble forecasts of the WRF model results had higher correlation and lower bias for lead times less than 36 h (Table 6).

The discharge error measurement indicated that longer lead times had lower accuracy as indicated by the increasing RMSE values. Accuracy assessments of lead time variation demonstrated that lead time dependency was almost negligible below the 36 h lead time in the 2002, 2011, and 2007 events (Fig. 12).

## 4.6 Time series analysis

As previously described, the spatial resolution of 8 km, the temporal resolution of 60 min, and the lead time of 36 h were

chosen for coupling the SURR and WRF models in the Imjin basin. Therefore, the time series analysis and the plots of forecast streamflow were created for the abovementioned spatial resolutions, temporal resolutions, and lead times. The accuracy of a streamflow forecast system is dependent upon how well the coupled models are able to make precise results. Consequently, to refuse or agree with the qualification of the model results, it is vital to establish accuracy measurements. The accuracy of a prediction is evaluated by comparing the observed, simulated and forecasted values. The simulation

estimates conducted with the observation MAP and MAE are illustrated as the SURR model input; however, the forecast obtained with the ultra-fine scale, real-time meteorological data from the WRF model are used as the input to drive the





SURR model. For the accuracy assessment of the coupled system, statistical error measures are used to describe the average deviations and compare the skill of hydrologic simulations and forecasts produced from the different inputs with observed streamflow. There are some efficiency criteria, such as NSE, MRE and REV, which are frequently used in hydrologic modeling assessments. The Jeonkok has no observation data for the 2002 event. The results of error measurements for the
2002, 2007 and 2011 events illustrated that the coupling system of the two models caused a decrease in the NSE and an increase in the error measurement indexes before and after linking the models (Table 7).

Within the NSE range set between 1 (i.e., the ideal value) and negative infinity, values lower than zero indicate that the mean value of the observed streamflow could have better estimate than the model provides. According to the calibration and verification of the SURR model, the results of the streamflow simulations are reasonable and stable with NSEs close to 1;
however, for the coupled SURR-WRF model, the NSE decreased dramatically. Here, it should be noted that the calibration of the SURR model parameters is done based on the rain gauge data and observed streamflow, while for real-time flood forecasting by the SURR-WRF coupled system, the real-time precipitation is forecasted using the WRF model. Therefore, the significant differences in NSE and the increases in the error measurement indexes for the SURR and coupled SURR-WRF models are related to the various sources of precipitation used as inputs for the hydrological model.

The performances of the SURR model in simulating the streamflow along with the SURR-WRF coupled model in forecasting the streamflow at the Gunnam, Jeonkok and Jeogseong stations in the 2002, 2007 and 2011 events are presented in Figure 13. The observed stream flow (black curve) is drawn to show the SURR model verification. The red curve indicates the forecast stream flow in the coupled SURR-WRF model, while the solid black curve shows the simulated streamflow using the observed meteorological data. The coupled SURR-WRF model is composed of the observed
precipitation until the onset of the forecast time, and it continues using the WRF data to a 36 h forecast lead time. The combinations of observed and real-time WRF data are used to drive the hydrological model. The observed and real-time forecast precipitations are shown separately in the upper and lower panels of Figure 13, respectively. This procedure is repeated for the next 6 hours to the end of the forecast time. Due to space limitation, briefly, one stream flow forecast is shown with relevant precipitation to indicate the real-time forecast discharge variation over time.

Interestingly, considering the runoff, the amplitudes of the simulated and observed peaks are quite similar in the SURR model simulation, while the amplitudes forecasted by the WRF model are different. This can be explained as the result of the two sources of precipitation, which are basically different. The spatial and temporal variation in the rainfall characteristics were not captured well by the WRF model in the real-time forecast data. In general, it is shown that the NWP models forecast less intermittent precipitation than indicated by the observed precipitation rates. Due to the precipitation
parameterization, most of the schemes used in the NWP models have deviations in the forecasted runoff hydrographs with respect to timing and amplitude compared to the measured runoff. Typically, the forecast floods underestimated the peak floods, and the forecasted flood errors are related to the inaccuracies in the real-time forecasted rainfall. Considering all real-time forecast cases from the start of the forecasting time until the end of the forecasting time, on average, it can be concluded that hydrological forecasts based on meteorological model inputs were able to reproduce the shape and the timing of the



calculated stream flow fairly well. However, the underestimation of the WRF model precipitation was noticeably affected by the real-time forecast discharge in all events.

## 5 Discussion

According to the recent studies on coupled hydro-meteorological models, it is necessary to diagnose and evaluate models to more robustly clarify where the models have weaknesses and need improvement. It is clear that the variations in spatial and temporal resolutions as well as in the lead time of the precipitation forecasts lead to notable differences in the accuracy of flood forecasting. However, it is not yet clarified at which spatial and temporal resolution and lead time the runoff is forecasted with rational accuracy. In this study, many analyses have been carried out to assess the ability of meteorological and hydrological models to provide hydrological predictions through the estimation of the errors associated with each model. The aim of the present study is to compare the different factors related to the coupling of rainfall runoff forecasting and meteorological systems; additionally, a real-time case study was used to quantify the accuracy assessment of each component. The results of real-time coupled SURR and WRF models highlighted the relative strengths and limitations of the models. The system accuracy assessment was composed of meteorological and hydrological model efficiency.

The results of point precipitation analysis and the spatial distribution of MAP generally indicated that the WRF model produced less accurate precipitation than the observed precipitation rates. The WRF model with high spatial resolution eliminated the error related to locating different stations in the same model grid cell with different observations. The point precipitation analysis showed that the skill of the WRF model varied considerably between rain gauge stations. This might be related to the precipitation parameterization schemes used in the WRF model. The catchment-scale assessment of the WRF model performance by MAP demonstrated the WRF model underestimated the MAP for the events. The hydrological model used in this study benefitted from lower RMSE values in MAP than in point precipitation.

The hypothesis that higher spatial and temporal resolution data have better accuracy is supported in this study; however, based on the findings, the temporal resolution did not have much of a negative effect on the inherent inaccuracy of the data. In general, the results indicated that the real-time forecasting system performed better with short forecasting lead times than longer ones. During this time, the effects of the initial conditions, spin up, regional characteristics and warm-up time were removed; thus, the results became more reliable. The coupled model performance for all events resulted in runoff peaks of coupled results in cases where time and height were in agreement; however, there was not a good fit with the simulated peaks compared to the observations. The results of the accuracy assessments indicated a decrease in NSE and an increase in error measurement indexes after linking the models. This could be related to the fact that the hydrological model calibration is done with rain gauge data and observed streamflow; however, the coupled SURR-WRF model used the real-time forecast rainfall. These are different sources of rainfall were used as input for the hydrological models. Therefore, it could be expected that the hydrological response in forecasting the streamflow would not match the simulated streamflow very well.



## 6 Conclusion and recommendations

The main conclusions of this study are listed below:

(1) The WRF model underestimated the precipitation in this study area in the point and catchment assessments.

(2) Comparing the results of the point and catchment scale indicated that the WRF model had better performance for the catchment-scale assessment. These findings led to the selection of the semi-distributed hydrological model.

(3) It was determined that spatial resolutions lower than 8 km did not affect the inherent inaccuracy of the flood forecasts in all events.

(4) The findings of the RMSE assessment for flood forecasting illustrated that variations in temporal resolution did not affect the RMSE significantly.

(5) The skill of the WRF model's real-time forecasts varied significantly with forecast lead time. Lead time variation demonstrated that lead time dependency was almost negligible below 36 h.

(6) It is concluded that real-time hydrological forecasts were able to reproduce the shape and the timing of the calculated streamflow. However, the underestimation of precipitation by the WRF model is noticeably affected by the real-time forecasted discharge in all events.

The promising results showed there is a vital need to improve accuracy, and this need should be addressed in future studies. The skill of the deterministic WRF model skill is not high enough to drive the SURR model with satisfactory results, and additional work should focus on improving flood forecasting using methods such as the extended Kalman filtering approach. In addition, the QPF is the most important factor driving the hydrological models in coupled studies; therefore, improvements that focus on the QPF post-processing are proposed. Since the lead time of forecasting is an important factor in real-time flood forecasting, future studies should also focus on potentially improving the lead time of flood forecasting.

## Acknowledgements

This research was supported by the National Research Foundation of Korea (NRF) grant funded by the Korean government (MSIP) (No. 2011-0030040).

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



**Table 1: SURR model sensitive parameters**

| Parameters | Description | Range | Unit |
|------------|-------------|-------|------|
| ALPHABF | Base flow coefficient | 0-2 | - |
| $K_{CH}$ | K coefficient in channel | 4000-10000 | - |
| $K_{SB}$ | K coefficient in subbasin | 30-60 | - |
| LAGSB | Lag time in subbasin | 1-4 | h |
| SURLAG | Surface runoff lag coefficient | 1.5-4 | $h^{-1}$ |

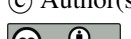



**Table 2: Statistical measures used to evaluate model performances**

| Index | Formula | Range | Ideal value |
|---|---|---|---|
| Nash-Sutcliffe Efficiency | $NSE = 1 - \dfrac{\sum_{i=1}^{N}(O_i - S_i)^2}{\sum_{i=1}^{N}(O_i - \bar{O})^2}$ | $(-\infty, 1)$ | 1 |
| Mean Relative Error | $MRE = \dfrac{1}{N}\sum_{i=1}^{N}\dfrac{S_i - O_i}{O_i}$ | $(-\infty, \infty)$ | 0 |
| Relative Error in Volume | $REV = \dfrac{\sum S_i - \sum O_i}{\sum O_i} \times 100$ | $(-\infty, \infty)$ | 0 |

$O_i$: Observed streamflow

$S_i$: Simulated streamflow

$\bar{O}$: Average of observed streamflow





**Table 3: Statistics of point precipitation analysis**

| Events | 2002 | | 2007 | | 2011 | |
|---|---|---|---|---|---|---|
| | Observation | WRF | Observation | WRF | Observation | WRF |
| ∑ (mm) | 10125 | 5813 | 32129 | 25446 | 33516 | 11818 |
| Min (mm) | 179 | 98 | 278 | 247 | 233 | 127 |
| Max (mm) | 405 | 282 | 907 | 599 | 790 | 253 |
| Underestimation (%) | 94.0 | | 84.8 | | 100 | |



**Table 4: Statistics of MAP analysis for 38 subbasins**

| Events | 2002 | | 2007 | | 2011 | |
|---|---|---|---|---|---|---|
| | Observation | WRF | Observation | WRF | Observation | WRF |
| $\sum$ (mm) | 11100 | 9146 | 19041 | 15263 | 17445 | 6710 |
| Min (mm) | 197 | 188 | 299 | 159 | 293 | 53 |
| Max (mm) | 351 | 349 | 642 | 494 | 743 | 219 |
| Underestimation (%) | 84.2 | | 78.9 | | 97.4 | |

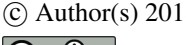



**Table 5: Precipitation assessment for individual and mean forecast real-time data**

|  |  |  | Event | | |
|---|---|---|---|---|---|
|  |  |  | 2002 | 2007 | 2011 |
| Individual forecast | Point assessment | RMSE | 84.49 | 212.80 | 91.53 |
|  | MAP assessment | RMSE | 59.67 | 160.48 | 68.49 |
|  | - | Error reduction (%) | 29.38 | 24.59 | 25.17 |
| Mean forecast | Point assessment | RMSE | 150.42 | 169.52 | 355.39 |
|  | MAP assessment | RMSE | 121.67 | 158.80 | 303.58 |
|  | - | Error reduction (%) | 19.11 | 6.32 | 14.59 |

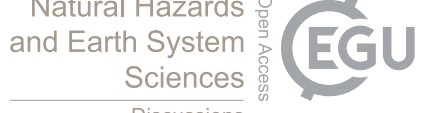



**Table 6: The relative bias and correlation assessments for different lead times**

| Lead time | Event 2002 | | Event 2007 | | Event 2011 | |
|---|---|---|---|---|---|---|
| (h) | Relative bias | Correlation | Relative bias | Correlation | Relative bias | Correlation |
| 0-12 | 65.00 | 0.11 | 32.00 | 0.17 | 54.00 | 0.04 |
| 13-24 | 34.00 | 0.16 | 22.00 | 0.42 | 52.00 | 0.27 |
| 25-36 | 60.00 | 0.20 | 27.00 | 0.37 | 59.00 | 0.38 |
| 37-48 | 67.00 | 0.15 | 33.00 | 0.35 | 61.00 | 0.18 |
| 49-60 | 76.00 | 0.12 | 40.00 | 0.30 | 64.00 | 0.17 |
| 61-72 | 93.00 | 0.03 | 43.00 | 0.11 | 78.00 | 0.09 |





**Table 7: Results of statistical error measurements in Imjin basin**

| Index | Gunnam Station | | Jeonkok Station | | Jeogseong Station | |
|---|---|---|---|---|---|---|
| | SURR | SURR-WRF | SURR | SURR-WRF | SURR | SURR-WRF |
| Event 2002 | | | | | | |
| NSE | 0.26 | -18.00 | - | - | 0.68 | -19.84 |
| MRE | -0.09 | 0.95 | - | - | -0.25 | 0.80 |
| REV | 0.16 | 0.70 | - | - | 0.03 | 0.53 |
| Event 2007 | | | | | | |
| NSE | 0.69 | -23.07 | 0.78 | -25.00 | 0.71 | -10.00 |
| MRE | -0.08 | -0.25 | -0.06 | -0.16 | -0.02 | -0.40 |
| REV | -0.18 | -0.29 | -0.12 | -0.23 | -0.09 | -0.41 |
| Event 2011 | | | | | | |
| NSE | 0.80 | -0.47 | 0.81 | -0.87 | 0.90 | -1.05 |
| MRE | -0.49 | -0.59 | -0.34 | -0.73 | -0.06 | -0.6 |
| REV | -0.08 | -0.54 | -0.63 | -0.67 | -0.45 | -0.56 |





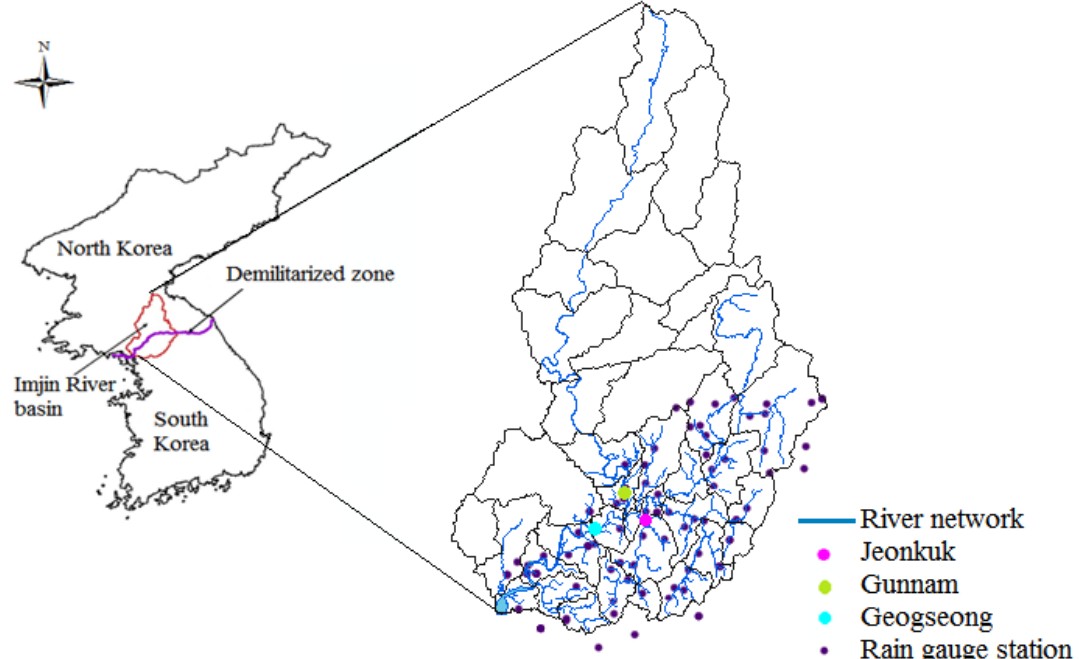

Figure 1: Location, subbasins, network, and water level and rain gauge stations of the Imjin River basin





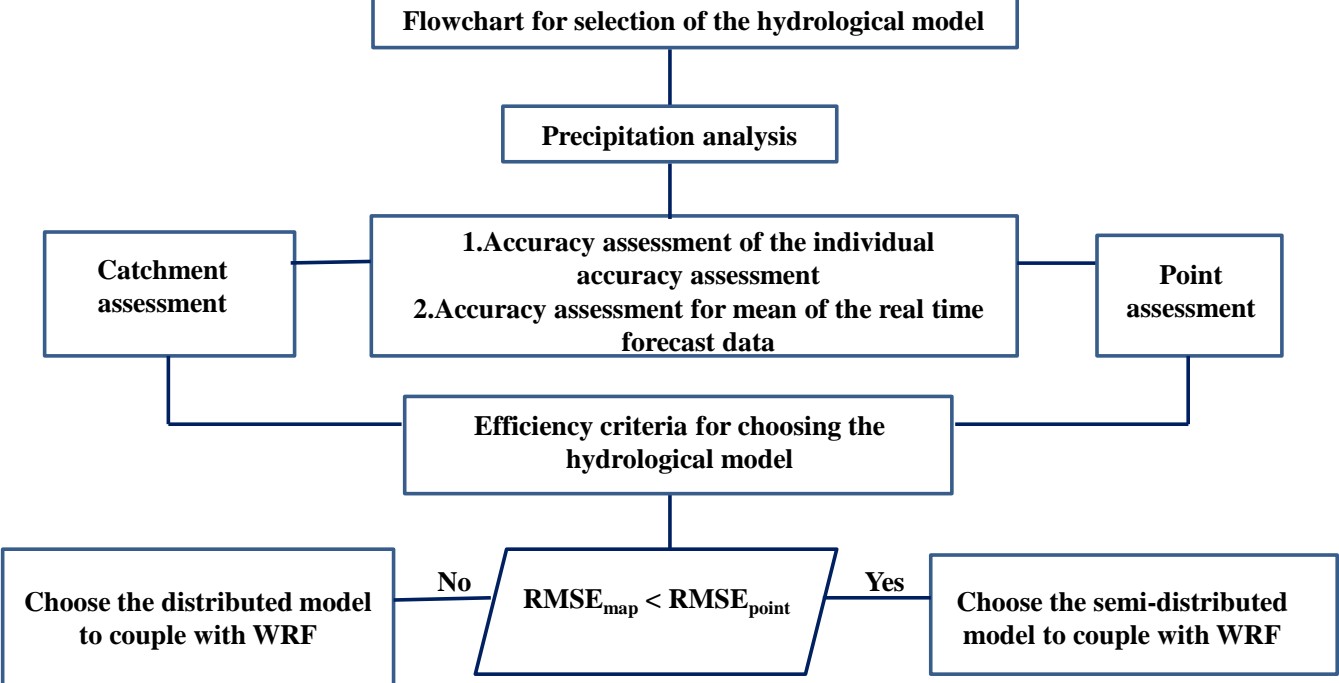

Figure 2: The flowchart of the hydrological model selection procedure





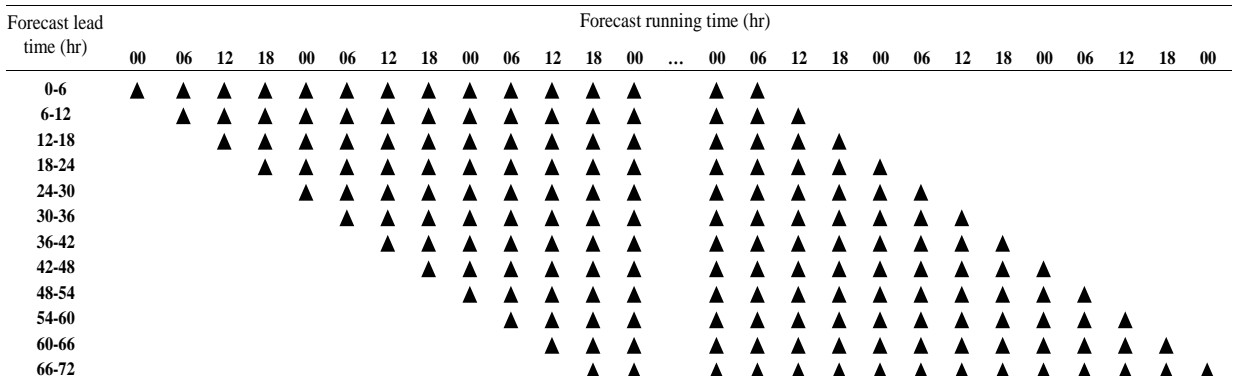

**Figure 3: The schematic construction of real-time forecast of the WRF model**



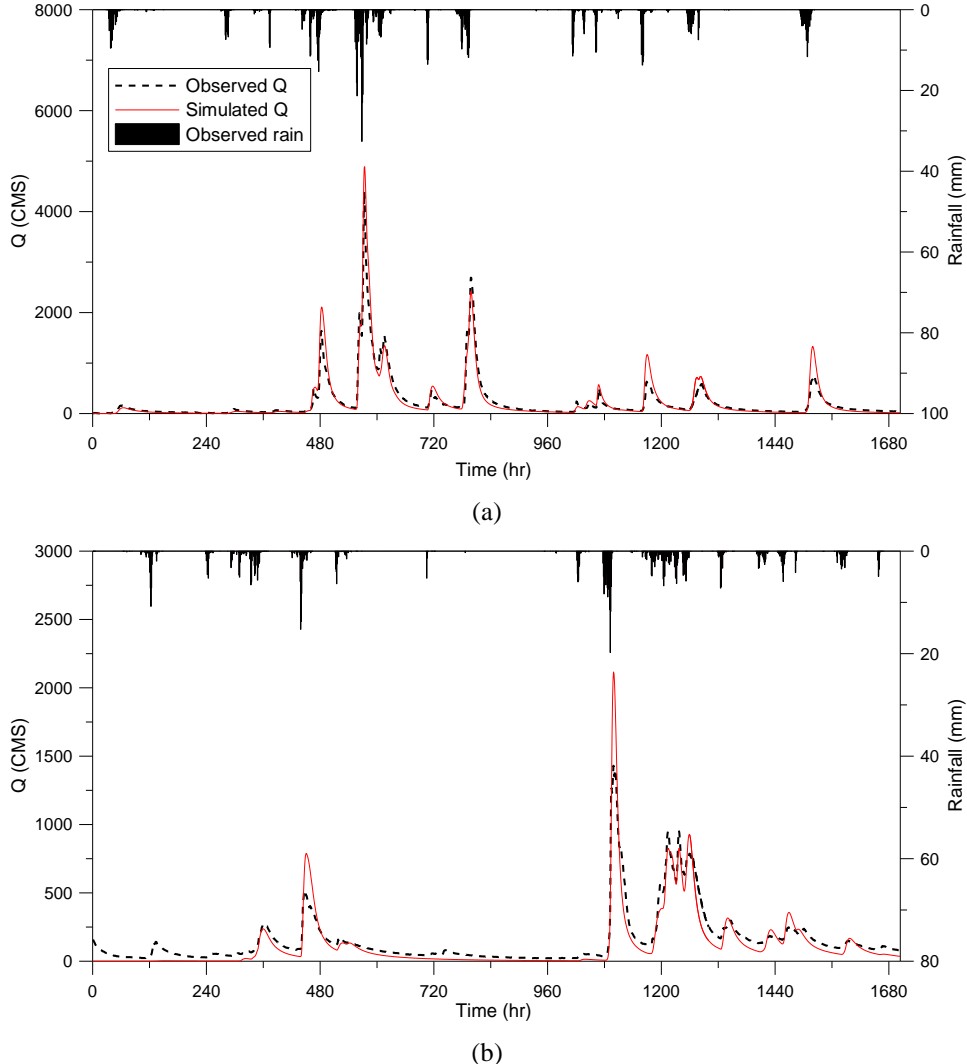

Figure 4: The results of the (a) calibration (event 2008) and (b) verification (event 2012) of the SURR model




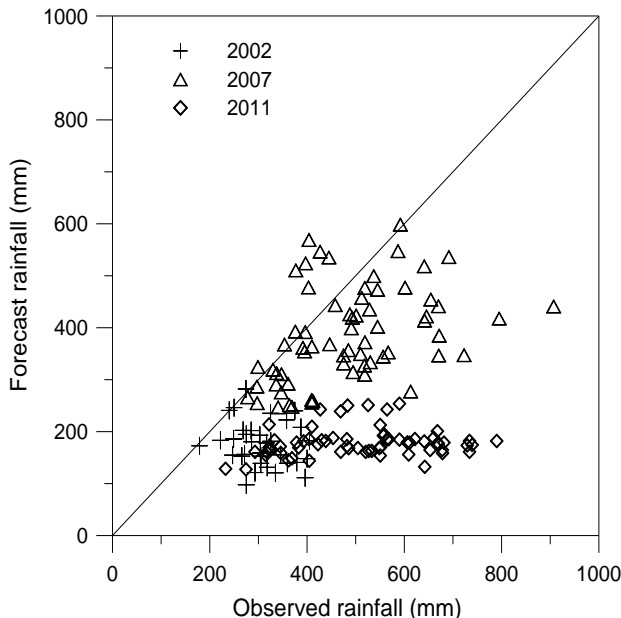

**Figure 5: Scatter plot of observed and forecast precipitation for the events of 2002, 2007 and 2011**



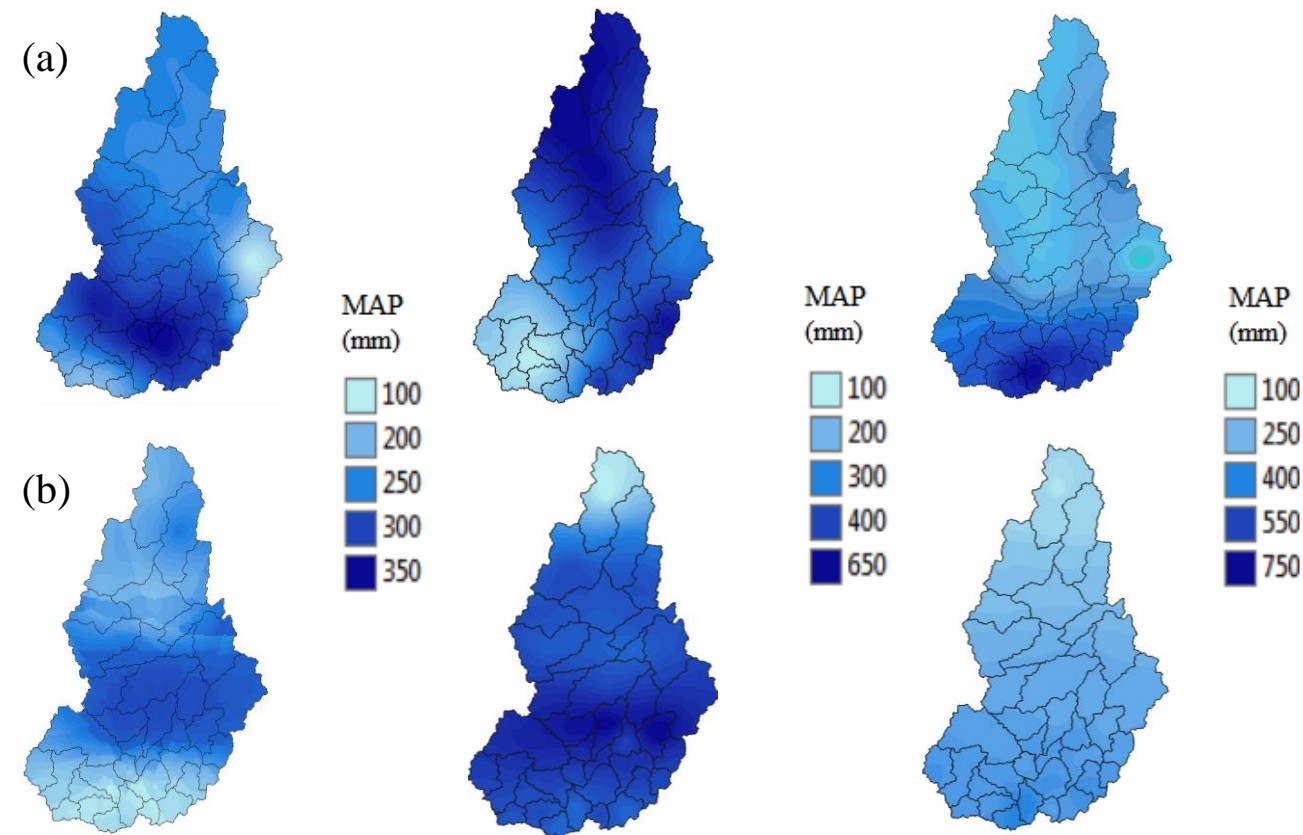

**Figure 6: Accumulated MAP in the Imjin basin for the 2002, 2007 and 2011 events (i.e., from left to right); (a) observation and (b) WRF model forecast precipitation**





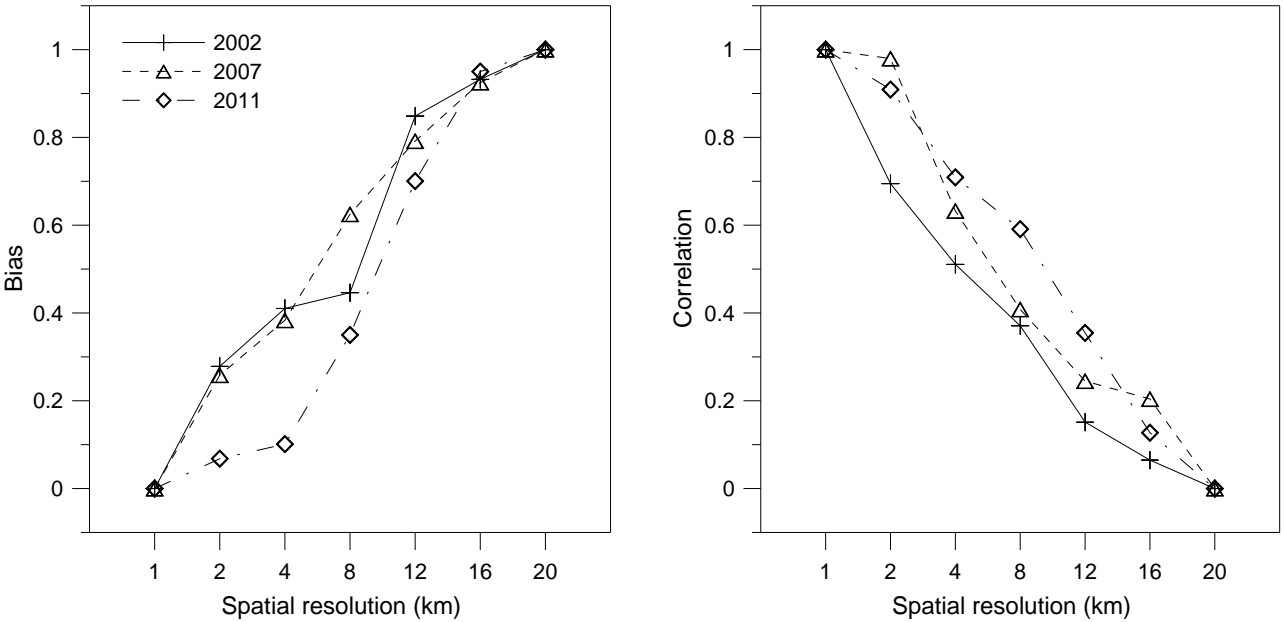

**Figure 7: The bias and correlation assessment for the different spatial resolutions in events 2002, 2007 and 2011**





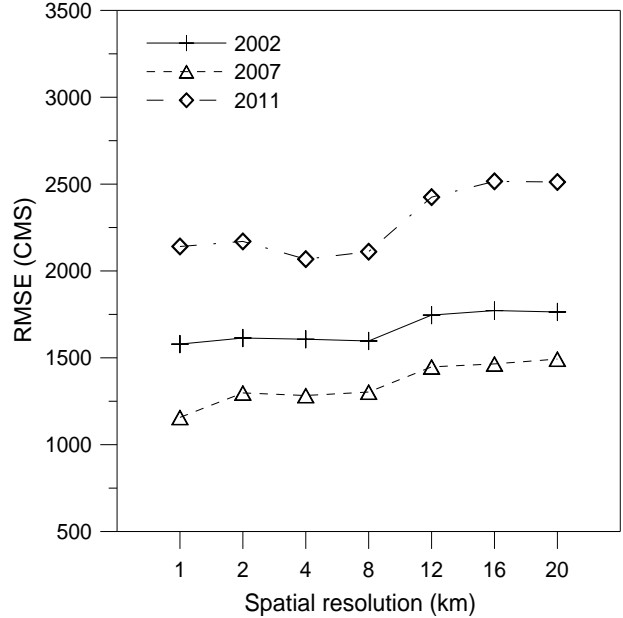

**Figure 8: Comparison of forecast flow RMSE for different spatial resolutions in events 2002, 2007 and 2011**



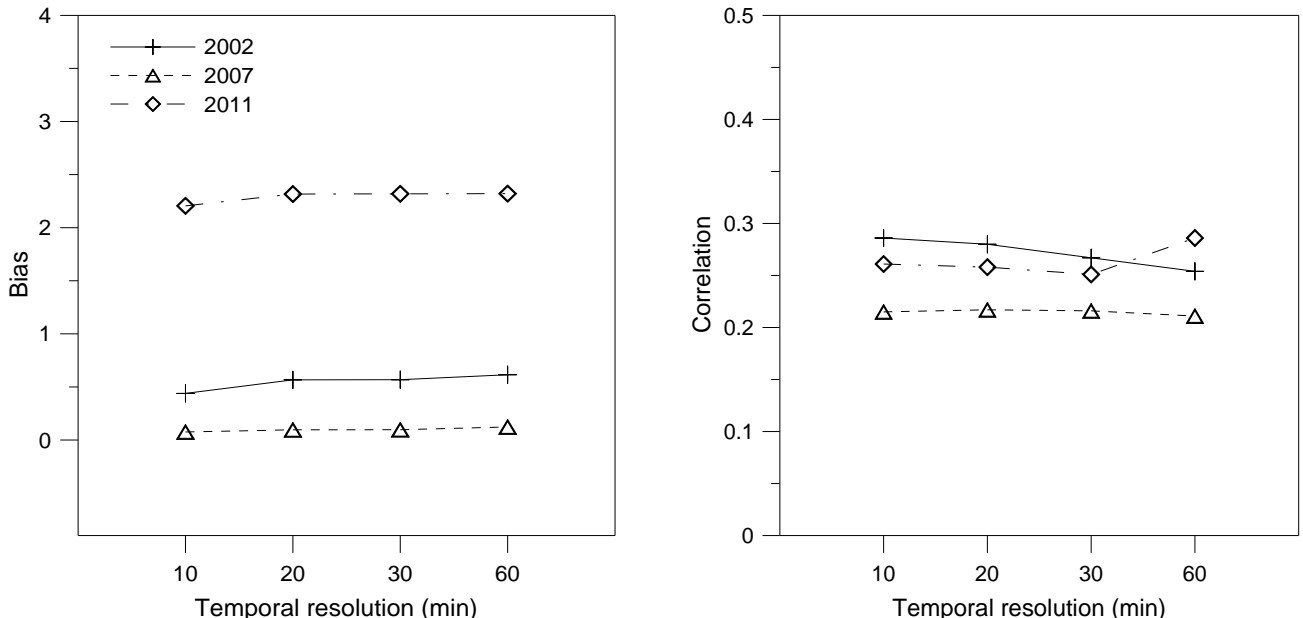

**Figure 9: The bias and correlation assessment for the different temporal resolutions of events 2002, 2007 and 2011**





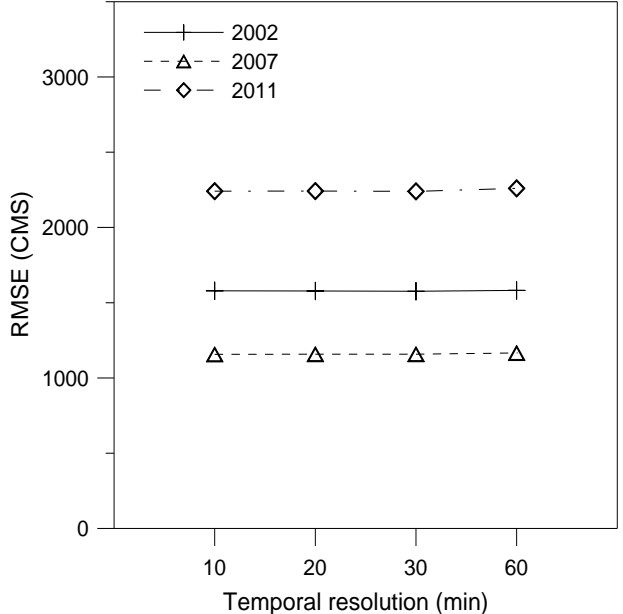

**Figure 10: Error measurement of different temporal resolutions of events 2002, 2007 and 2011**







**Figure 11: Observed and forecasted rainfall for lead times (a) 0-12h, (b) 13-24h (c) 12-36h (d) 37-48h,(e) 49-60h, (f) 61-72h of all the events**




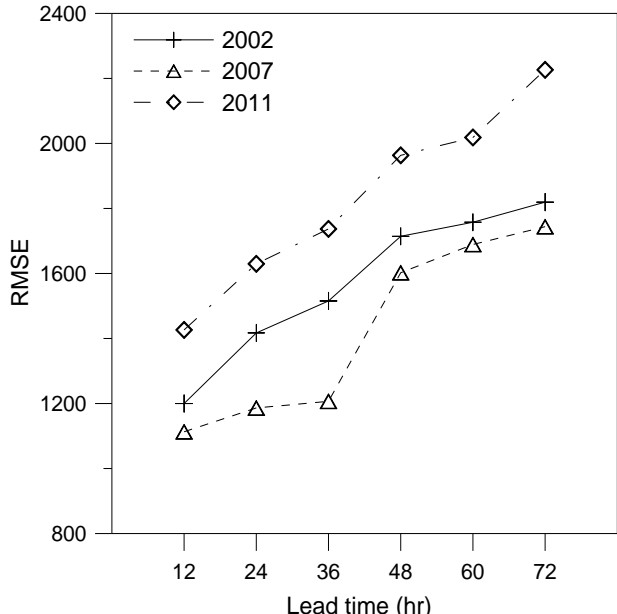

**Figure 12: Comparison of RMSE of forecast flow for different lead times in events 2002, 2007 and 2011**





(a) Event 2002





**Figure 13: Comparison of simulated, observed and forecasted flow for (a) 2002, (b) 2007 and (c) 2011 events**