# Peer review of "Accuracy assessment of real-time flood forecasting of coupled hydrological and mesoscale meteorological models"

_Natural Hazards and Earth System Sciences, 2017_

## Referee Comment (RC1) · Anonymous Referee #1 · 20 Mar 2018

**1. General comments**

This manuscript describes the real time forecasting using coupled hydrological and mesoscale meteorological models for Imjin transboundary river basin in Korea. The authors analyzed to get optimal temporal and spatial resolution of Weather Research Forecasting (WRF) model and hydrologic model. This manuscript provides optimal resolution of the mesoscale and hydrologic models for the applied area. But there is no scientific new findings or procedure. And also lack of description for fundamental data used. The measured rainfall in this area, especially transboundary basin, is not clear and enough. For example, 2/3 of area is placed in North Korea, did the authors

analyzed the hourly rainfall of North Korea? If the rainfall data of North Korea was not used, the results of this research are not verified for the whole Imjin river basin including rainfall and streamflow. If the meteorological and hydrological data were not sufficiently used, the procedure and accuracy assessment for the coupling hydrological and meteorological model have no strong basement. Also, only three big flood events were considered. Is it enough?

2. Specific Comments:

* page 3, line 1, ultra-fine - Is there any criteria for the ultra-fine?, Reference? * page 3, line 10, seventh largest river in Korea - South or North or both? * page 3, line 14, 1100 mm - need reference for the description * page 3, line 17, it was difficult∼ - How did the authors handle the data for the North Korea? * page 3, line 20-23, The political issue is not necessary. * page 3, line 26-30, event 20020828-0904 - numbering for the floods are not general, recommend tables and concise numbering. * page 3, line 30- - Is there severe damages in Imjin river? * page 4, line 3, different nature - what the authors think the different natures between North and South Korea in Imjin River * page 4, line 5, the number of rain gauges is changed, why? Need the rain gauges on the Fig. 1. * page 5, line 13, SURR semi-distributed continuous rainfall runoff model - how did the authors consider spatial resolution using SURR model and actual evapotranspiration for the continuous rainfall runoff simulation. In the manuscript, there is no mention about the evapotranspiration, even though the event applied for two months period (2012701-0910). * page 5, line 28∼, calibration for the hydrologic model - it is not enough for the calibration analysis, more description is necessary for parameters for each event and statistic criteria. Spatial resolution used for the hydrologic model is not clear (km x km or 38 subbasins?) * page 12, line 12-17, it is not necessary for this manuscript. * page 17, table 1, add bias and correlation, RMSE * page 18, table 3, additional figure for the statistics is more efficient for the readers * page 1, figure 1, location of rain gauges is necessary * page 28, figure 6, legend should be used same scale * page 33, figure 11, Y-axis subtitle is misspelled. * page 12, figure 12, unit is missed in Y-axis

---

## Referee Comment (RC2) · Anonymous Referee #2 · 19 Apr 2018

The paper describes evaluation of WRF-downscaled driving data used in a hydrological model in a catchment on the border between North and South Korea. The paper sets out to analyse the optimal setting of temporal and spatial resolution of the precipitation modelling to produce a good hydro-meteorological forecast for the area. However, the analysis is only done over three case studies and the results do not support the conclusions. The sample size is simply too small. Further, the paper is not well organised and I am often confused by the description of the methods and results; and there is not enough relevant references. I therefore do not recommend the paper to be published in its current form.

Specific comments.

1. As stated already the paper would be more interesting if the study were done over a longer time period to support a robust statistical analysis of the results. It is OK to highlight certain features through case studies, but you cannot build your results on only three cases.

2. The text is often too long and some details described to carefully. I also sometimes struggle to understand what the authors mean, so the paper would need a thorough language revision.

3. There is a lack of relevant references to support the statements made in the introduction, and the references are often too old.

---

## Author Comment (AC1) · 29 May 2018

The authors appreciate the careful review and constructive suggestions and thank the reviewers for the effort and time put into the review of the manuscript. Each comment has been carefully considered and responded in italic format. It is our belief that the manuscript is substantially improved after making the suggested edits.

Since our response file includes figures and tables, we uploaded the Author's responses to referee1 as a .pdf file as a supplement to the comments.

Thank you for your review of our paper. We have answered each of your points in italic

format.

Please also note the supplement to this comment:
https://www.nat-hazards-earth-syst-sci-discuss.net/nhess-2017-447/nhess-2017-447-AC1-supplement.pdf
* * *
[Figure]

**Supplement:**

**Response to Referee #1:**

*Thank you for your review of our paper. We have answered each of your points below in italic format.*

15 **1. General comments**

This manuscript describes the real time forecasting using coupled hydrological and mesoscale meteorological models for Imjin transboundary river basin in Korea. The authors analyzed to get optimal temporal and spatial resolution of Weather Research Forecasting (WRF) model and hydrologic model. This manuscript provides optimal resolution of the mesoscale and hydrologic models for the applied area. But there is no scientific new

20 findings or procedure.

➤ *This paper has two objectives. The first objective is to find proper hydrological model to couple with a meteorological model. Evaluation of point precipitation and the Mean Areal Precipitation (MAP) could lead to robust decision making in the distributed (which uses the point precipitation data as input) and semi-*

25 *distributed (which uses the MAP as input) hydrological models. In order to find the proper hydrological model, the assessment of the forecasted precipitation for rain gauge stations and MAP are done using individual forecasts and the mean of the forecast data. In individual forecast analysis, the quality of forecast precipitation is analyzed by comparing the values with observation data. For evaluating the mean of forecast data, analysis is done by the average ensemble method using equal weighting to the members, which are*

30 *lagged by 6 hours. This approach included forecast averages from multiple lead times, which were then compared with observed data. The Root Mean Square Error (RMSE) is used for the comparison between real-time forecast data and the observed data. By comparing the RMSE in point and catchment scale we can select the proper hydrological model. The following flowchart shows the procedure of the proper hydrological model selection.*

[Figure]

> *According to our findings the point scale precipitation assessment had higher RMSE than the catchment scale precipitation assessment. The results are shown as following.*

| | | | Event | | |
|---|---|---|---|---|---|
| | | | **2002** | **2007** | **2011** |
| **Individual forecast** | **Point assessment** | **RMSE** | 84.49 | 212.80 | 91.53 |
| | **MAP assessment** | **RMSE** | 59.67 | 160.48 | 68.49 |
| | **-** | **Error reduction (%)** | 29.38 | 24.59 | 25.17 |
| **Mean forecast** | **Point assessment** | **RMSE** | 150.42 | 169.52 | 355.39 |
| | **MAP assessment** | **RMSE** | 121.67 | 158.80 | 303.58 |
| | **-** | **Error reduction (%)** | 19.11 | 6.32 | 14.59 |

> *As a result of the lower RMSE in MAP than that from the point precipitation accuracy assessment, the semi-distributed hydrological model may be a better choice for this study. In coupled hydro-meteorological studies there is a lack of literature for choosing proper hydrological model (lumped, semi- and fully-distributed) based on the meteorological model results. The evaluation of the real-time precipitation in point and catchment scale provided new findings for choosing the proper hydrological model to couple with the meteorological model. This procedure was not already known or considered in previous studies.*

> *The second objective of this study is to evaluate the effects of lead time, spatial and temporal resolution variation of the WRF model data on the performance of coupled hydro-meteorological models. In previous coupled studies the temporal and spatial resolution of the hydrological models is much finer than the meteorological models. However, in this study the high resolution spatial and temporal resolution of the meteorological model gave us a chance to overcome the limitation of previous studies. Another contribution of the paper is that we propose a framework for evaluating the precipitation and discharge accuracy variation for different spatial and temporal resolution and forecast lead-time of the meteorological model in a real-time coupled hydro-meteorological study. It is already known that higher resolution modeling leads to*

*more accurate forecasts. Therefore, the results should indicate that, by decreasing the spatial resolution, the bias increases, and the correlation coefficient decreases. The MAP bias variation showed that after 8km spatial resolution the bias increased significantly in all events. The MAP correlation assessment indicated that after 8km spatial resolution the correlation decreased significantly. The RMSE assessment of the*
5 *forecast discharge showed that the RMSE increased after 8km spatial resolution. Therefore, it is shown that the spatial resolutions lower than 8 km did not affect the inherent inaccuracy of the flood forecasts in all events, while after that, the error increased to a higher level for all events the results of the temporal resolution did not show the variation by increasing the temporal resolution. For forecast lead-time evaluation, it is also known that the forecast skill decreases with the increase in lead time, which is related to*
10 *the higher uncertainty in the forecast data. In this part different lead-time intervals are chosen to show the over- or underestimation of the WRF data comparing with the observed precipitation. The discharge error measurement indicated that longer lead times had lower accuracy as indicated by the increasing RMSE values. Accuracy assessments of lead time variation demonstrated that lead time dependency was almost negligible below the 36 hr lead time in all events. Our finding can be summarized as following tables.*

15

| | *Spatial Resolution* | | | *Temporal resolution* | | |
|---|---|---|---|---|---|---|
| | *2002* | *2007* | *2011* | *2002* | *2007* | *2011* |
| *MAP bias* | *8km* | *8km* | *8km* | *No significant changes* | *No significant changes* | *No significant changes* |
| *MAP correlation* | *8km* | *8km* | *8km* | *No significant changes* | *No significant changes* | *No significant changes* |
| *Q RMSE* | *8km* | *8km* | *8km* | *No significant changes* | *No significant changes* | *No significant changes* |

| | *Forecast led-time* | | |
|---|---|---|---|
| | *2002* | *2007* | *2011* |
| *Q RMSE* | *36hr* | *36hr* | *36hr* |

And also lack of description for fundamental data used. The measured rainfall in this area, especially transboundary basin, is not clear and enough. For example, 2/3 of area is placed in North Korea, did the authors
20 analyzed the hourly rainfall of North Korea? If the rainfall data of North Korea was not used, the results of this research are not verified for the whole Imjin river basin including rainfall and streamflow. If the meteorological and hydrological data were not sufficiently used, the procedure and accuracy assessment for the coupling hydrological and meteorological model have no strong basement.

25 ➢ *In this study the meteorological data includes rainfall, temperature, wind speed, relative humidity and solar radiation which used as the input data of our semi-distributed hydrological model (SURR model). We have two types of the meteorological data.*
➢ *The first one is the observed data which is forced to the SURR model to simulate the streamflow. The second one is the real-time forecast data which is provided by WRF model and is used to forecast the streamflow in*
30 *Imjin basin. The Imjin basin is divided into 38 sub-basins and we used the observed and forecast data to get*

*the MAP and MAE for SURR model. First, in order to get the observed MAPs and MAEs we use the observed data from rain gauge and meteorological stations which are located in South Korea. Therefore, by using the rain gauge data and the Thiesen polygons using GIS, the observed MAPs and MAEs are estimated for the whole catchment. For calibration and verification of the SURR model we use the observed meteorological data and observed streamflow. The observed streamflow stations are also located in South Korea (Gunnam, Jeonkuk and Jeogseong stations). In calibration of the SURR model we have two types of the model parameters, which are subjective and objective parameters. The subjective parameters can be estimated based on basin characteristics using GIS while the objective parameters are computed in model calibration process. In order to get the subjective parameters we used the information from the whole catchment. For obtaining the objective parameters we use the calibration procedure and observed rainfall (to get the MAP and AME) and observed discharge (In Gunnam, Jeonkuk and Jeogseong stations).*

➢ *Second, in order to get the forecast MAPs and MAEs we use the real-time WRF forecast data. The WRF model covers the whole catchment with high spatial resolution (1km × 1km). Therefore, by using the real-time forecast meteorological data the forecast MAPs and MAEs are estimated by spatially interpolation of the Thiessen polygons for each sub-basin of the whole catchment. Here we use the forecast data of North and South Korea to obtain our input for the SURR model and run the SURR model to get the real-time flood forecasts. This data is used to forecast the streamflow in Imjin basin. In order to run the SURR model for runoff simulation and forecast, the observed and forecast MAPs of the whole catchment are used respectively.*

Also, only three big flood events were considered. Is it enough?

➢ *The number of events used in hydro-meteorological studies strongly depends on the purpose of the study. In this study we have two objectives. The first objective is grounded on the idea to find the proper type of the hydrological model to couple with a meteorological model for a real-time flood forecasting. The results of this part led to choose the semi-distributed hydrological model. We totally used nine events in this study, six events for the calibration and verification of the hydrological model and three events used for the real-time flood forecasting by coupling SURR-WRF model. The results of the comparing the point scale and catchment scale of the precipitation analysis supports our first objective to find the proper hydrological model. In order to follow our first objective the number of events provided the required information to get a judgment for choosing the proper hydrological model.*

➢ *The second objective is to evaluate the effects of lead time, spatial and temporal resolution variation of the WRF model data on the performance of coupled hydro-meteorological models. For our second objective the real-time forecast accuracy variation assessment is done by considering the effects of the lead time forecast, spatial and temporal resolution of the meteorological model. We used three flood events which are the most important floods for the study area. We considered these events to have a reasonable chance of seeing such that effects and to provide the required sample size for comparative analysis for this research. The precipitation analysis and discharge evaluation for different spatial resolutions showed that by decreasing the spatial resolution the accuracy of the forecast decreased (following figures). Therefore it can be found that by increasing the number of events this trend will not change.*

[Figure]

[Figure]

[Figure]

> *Also temporal resolution variation showed that the forecast accuracy in precipitation and discharge analysis did not change. Therefore increasing the number of events will not change this trend of accuracy variation.*

[Figure]

[Figure]

[Figure]

> *Also for the forecast lead-time assessment, the forecast skill decreases with the increase in lead time (following figure) and increasing the number of events will follow this trend too.*

[Figure]

> *These events are considered to support the second objective and proposed analysis of our research. The number of three events is considered for comparison of the results and this number provides the specific aims of the analysis.*

**2. Specific Comments:**

* page 3, line 1, ultra-fine - Is there any criteria for the ultra-fine?, Reference?

> *There is no criterion for choosing the ultra-fine scale. However in meteorological models the 1km horizontal resolution is 10 times finer than the usual mesoscale models (10 km resolution). In order to make a clear orientation, we changed the "ultra-fine scale" to "high resolution" in the manuscript.*

* page 3, line 10, seventh largest river in Korea - South or North or both?

> *Imjin basin is the 7$^{th}$ largest river in Korean peninsula (North and South Korea).*

* page 3, line 14, 1100 mm - need reference for the description

> *We modified this part by adding reference for annual precipitation in Imjin basin.*

* page 3, line 17, it was difficult_ - How did the authors handle the data for the North Korea?

> *In this study the meteorological data includes rainfall, temperature, wind speed, relative humidity and solar radiation which used as the input data of our semi-distributed hydrological model (SURR model). We have two types of the meteorological data.*

> *The first one is the observed data which is forced to the SURR model to simulate the streamflow. The second one is the real-time forecast data which is provided by WRF model and is used to forecast the streamflow in Imjin basin. The Imjin basin is divided into 38 sub-basins and we used the observed and forecast data to get the MAP and MAE for our hydrological model.*

> *First, in order to get the observed MAPs and MAEs we use the observed data from rain gauge and meteorological stations which are located in South Korea. Therefore, by using the rain gauge data and the Thiesen polygons using GIS, the observed MAPs and MAEs are estimated for the whole catchment. The MAPs and MAEs are estimated by spatially interpolation of the Thiessen polygons for each sub-basin. For calibration and verification of the SURR model we use the observed meteorological data and observed streamflow. The observed streamflow stations are also located in South Korea (Gunnam, Jeonkuk and Jeogseong stations). In calibration of the SURR model we have two types of the model parameters, which are subjective and objective parameters. The subjective parameters can be estimated based on basin characteristics using GIS while the objective parameters are computed in model calibration process. In order to get the subjective parameters we use the information from the whole catchment. For adjusting the objective parameters we use the calibration procedure and observed rainfall (to get the MAP and AME) and observed discharge (In Gunnam, Jeonkuk and Jeogseong stations).*

> *Second, in order to get the forecast MAPs and MAEs we use the real-time WRF forecast data. The WRF model covers the whole catchment with high spatial resolution (1km × 1km). Therefore, by using the real-time forecast meteorological data and the Thiesen polygons using GIS, the forecast MAPs and MAEs are estimated for the whole catchment. The forecast MAPs and MAEs are estimated by spatially interpolation of the Thiessen polygons for each sub-basin. Here we use the forecast data of North and South Korea to obtain our input for the SURR model and run the SURR model to get the real-time flood forecasts.*

\* page 3, line 20-23,  The political issue is not necessary

> *We agree with the reviewer's opinion. The sentences (lines 20-23) are removed as you suggested.*

\* page 3, line 26-30, event 20020828-0904 - numbering for the floods are not general, recommend tables and concise numbering.

> *We modified the numbering for the floods in the following way.*

*Table 1: list of the investigated events in Imjin basin*

| Case number | Event ID | Event period |
| --- | --- | --- |
| 1 | 2002 | August 28 – September 4, 2002 |
| 2 | 2007 | July 23 – September 4, 2007 |
| 3 | 2011 | July 25 – July 30, 2011 |

\* page 3, line 30- - Is there severe damages in Imjin river?

> *The three flood events caused sever damages in vast area of Korean peninsula including the Imjin basin. The flood events in Imjin basin caused damages to the buildings, agricultural fields, roads, water structures, military equipment and marine facilities. In Imjin basin the number of death is 41, 4 and 37; number of the property damages is 14, 22 and 95 and total financial damages are approximately 3500000, 800000 and 1400000\$ in the events 2002, 2007 and 2011 respectively (MOIS, 2002; 2007 and 2011).*

\* page 4, line 3, different nature - what the authors think the different natures between North and South Korea in Imjin River

> *In Imjin basin for North Korea there are more mountains and higher altitudes by comparing with South Korea (fig.1d). Therefore the North and South Korea have different natures in Imjin basin. The differences between North and South Korea for average, maximum and minimum temperature, precipitation and relative humidity are shown in the following table. It should be noted that this data are provided by Global Telecommunication System (GTS) for South and North Korea (KDI, 2003).*

| Meteorological data | South Korea | North Korea |
|---|---|---|
| Average temperature (°C) | 12.1 | 8.7 |
| Maximum temperature (°C) | 38.4 | 22.6 |
| Minimum temperature (°C) | -20.2 | -16.7 |
| Precipitation (mm) | 1361.8 | 1173.2 |
| Relative humidity (%) | 67.5 | 76.0 |

\* page 4, line 5, the number of rain gauges is changed, why? Need the rain gauges on the Fig. 1.

➢ *Yes, there are 33 rain gauge stations for the event 2002 and 66 stations for the events 2007 and 2011. The number of rain gauges is changed after 2002 due to the Gunnam flood control project which started in 2003. The figure 1 is modified and the rain gauge stations are added to the figure.*

\* page 5, line 13, SURR semi-distributed continuous rainfall runoff model - how did the authors consider spatial resolution using SURR model and actual evapotranspiration for the continuous rainfall runoff simulation. In the manuscript, there is no mention about the evapotranspiration, even though the event applied for two months period (2012701- 0910).

➢ *The SURR model requires the MAP and MAE for each sub-basin as input data. The effects of the different spatial resolutions of the WRF model are considered by calculating MAPs and MAEs with different spatial resolutions. Then the MAPs and MAEs are forced to the SURR model to assess the effect of the different spatial resolutions on the flood forecast. More details regarding the calculation of evapotranspiration added to the manuscript. The evapotranspiration is calculated using FAO Penman-Monteith (FAO PM) formula which is a standard method for estimating evapotranspiration (ET). The FAO PM method is as follows:*

$$ET = \frac{0.408\Delta(R_n - G) + \gamma\frac{900}{T+273}u_2(e_s - e_a)}{\Delta + \gamma(1 + 0.34u_2)}$$

➢ *where ET is the evapotranspiration [mm day$^{-1}$], $R_n$ is the net radiation at the crop surface [MJ m$^{-2}$ day$^{-1}$], G is the soil heat flux density, which is relatively small for daily and ten-day periods [MJ m$^{-2}$ day$^{-1}$], T is the air temperature at a height of 2 m [ ℃], $u_2$ is the wind speed at a height of 2 m [m s$^{-1}$], $e_s$ is the saturation vapor pressure [KPa], $e_a$ is the actual vapor pressure [KPa], $e_s$-$e_a$ is the saturation vapor pressure deficit [KPa], $\Delta$ is the slope vapor pressure curve [KPa ℃$^{-1}$], and γ is the psychrometric constant [KPa ℃$^{-1}$].*

➢ *The meteorological data are used to calculate the ET and then the Thiessen polygons are used by GIS to estimate the MAE for each sub-basin. The observed meteorological data including wind speed, relative humidity, temperature and solar radiation are used to calculate the observed MAE. Also the real-time forecast meteorological data from WRF model, wind speed, relative humidity, temperature and solar radiation are used to calculate the real-time forecast MAE. The observed MAP and MAE are used to simulate streamflow and the real-time forecast MAP and MAE are used to forecast streamflow in Imjin basin.*

* page 5, line 28_, calibration for the hydrologic model - it is not enough for the calibration analysis, more description is necessary for parameters for each event and statistic criteria. Spatial resolution used for the hydrologic model is not clear (km x km or 38 subbasins?)

5 ➤ *In the SURR model, there are two types of parameters, which are subjective and objective parameters. The subjective parameters can be estimated based on basin characteristics using GIS while the objective parameters are computed in model calibration process. The subjective and the objective parameters of the SURR model are presented in table 2. For rainfall runoff simulation, the sensitive parameters of the SURR model are $P_{ch}$, $P_{sb}$, $K_{sb}$ and $K_{ch}$.*

*Table 2: the subjective and objective parameters in SURR model*

| Subjective parameters | Definition | Unit | Estimation method |
| --- | --- | --- | --- |
| AKM | Subbasin area | $km^2$ | GIS |
| SLP | Mean slope of the subbasin | m/m | GIS |
| Z | Depth of soil layer | m | GIS |
| SAT | Rate of water content at saturation | mm/mm | GIS |
| FC | Rate of water content at field capacity | mm/mm | GIS |
| WP | Rate of water content at wilting point | mm/mm | GIS |
| KS | Saturated hydraulic conductivity | mm/h | GIS |
| CN2 | Runoff curve number under AMC II | - | GIS |
| **Objective parameters** | | | |
| LHILL | Mean slope length | m | Calibration |
| SURLAG | Surface runoff lag coefficient | h | Calibration |
| LAGSB | Lag time of the subbasin | h | Calibration |
| LATLAG | Lateral flow lag coefficient | h | Calibration |
| SEPLAG | Delay time for water percolating | h | Calibration |
| GWLAG | Delay time for aquifer recharge | h | Calibration |
| ALPHA_BF | Baseflow recession constant | - | Calibration |
| AQMIN | Threshold water level in shallow aquifer for baseflow | mm | Calibration |
| $K_{sb}$ | K coefficient of the subbasin | $h^{Psb}$ | Calibration |
| $P_{sb}$ | P coefficient of the subbasin | - | Calibration |
| $K_{ch}$ | K coefficient of the channel | $s^{Psb}$ | Calibration |
| $P_{ch}$ | P coefficient of the channel | - | Calibration |

➤ *The calibration and verification events used in this study are provided in table 3. The SURR model was calibrated for the Imjin basin using the observed rainfall and streamflow, and the optimized parameters*
15 *resulted in good agreement between the observed and simulated streamflow during the verification periods. The statistical analyses of the SURR model simulations for the calibration and verification events are shown in table 4. For the sake of the brevity, the results of the calibration and verification are shown for the 2008 and 2012 events for Jeonkuk station (Fig. 4). A detailed description of the SURR model is reported in Bae and Lee (2011).*

20

*Table 3: The calibration and verification periods for SURR model simulations*

| Case number | Event ID | Event period |
| --- | --- | --- |
| 1 | Calibration | July 23 – September 3, 2007 |
| 2 | Calibration | July 1 – August 22, 2008 |
| 3 | Verification | June 21 – August 4, 2009 |
| 4 | Verification | July 9 – August 20, 2010 |
| 5 | Verification | June 16 – August 2, 2011 |
| 6 | Verification | July 31 – September 13, 2012 |

*Table 4: statistical analysis for simulated discharge for calibration and verification periods in SURR model*

| | Calibration period
*July 23 – September 3, 2007* | | | Calibration period
*July 1 – August 22, 2008* | | | Verification period
*June 21 – August 4, 2009* | | |
| --- | --- | --- | --- | --- | --- | --- | --- | --- | --- |
| | Gunnam | Jeonkuk | Jeogseong | Gunnam | Jeonkuk | Jeogseong | Gunnam | Jeonkuk | Jeogseong |
| RMSE | 629.36 | 182.87 | 864.82 | 599.13 | 139.52 | 609.68 | 632.18 | 196.79 | 766.87 |
| Nash | 0.68 | 0.82 | 0.75 | 0.70 | 0.83 | 0.79 | 0.57 | 0.85 | 0.79 |
| Correlation | 0.85 | 0.95 | 0.91 | 0.82 | 0.97 | 0.93 | 0.84 | 0.96 | 0.92 |
| REV | -0.34 | -0.32 | -0.38 | 0.37 | 0.03 | 0.08 | 0.16 | -0.22 | 0.03 |
| | Verification period
*July 9 – August 20, 2010* | | | Verification period
*June 16 – August 2, 2011* | | | Verification period
*July 31 – September 13, 2012* | | |
| | Gunnam | Jeonkuk | Jeogseong | Gunnam | Jeonkuk | Jeogseong | Gunnam | Jeonkuk | Jeogseong |
| RMSE | 702.59 | 263.35 | 779.22 | 621.33 | 220.18 | 704.99 | 688.67 | 77.19 | 616.71 |
| Nash | 0.62 | 0.71 | 0.67 | 0.71 | 0.89 | 0.85 | 0.59 | 0.78 | 0.66 |
| Correlation | 0.63 | 0.92 | 0.84 | 0.84 | 0.97 | 0.93 | 0.62 | 0.95 | 0.81 |
| REV | 0.23 | -0.34 | -0.07 | -0.09 | -0.19 | -0.11 | -0.28 | -0.20 | -0.05 |

[Figure]

[Figure]

[Figure]

Event 2008

[Figure]

Event 2012

* page 12, line 12-17, it is not necessary for this manuscript.

➢ *The sentences (lines 20-23) are removed as you suggested.*

* page 17, table 1, add bias and correlation, RMSE

➢ *The bias and RMSE equations are added to the table 5.*

| Index | Formula | Range | Ideal value |
|---|---|---|---|
| Nash-Sutcliffe Efficiency | $NSE = 1 - \dfrac{\sum_{i=1}^{N}(O_i - S_i)^2}{\sum_{i=1}^{N}(O_i - \bar{O})^2}$ | $(-\infty, 1)$ | 1 |
| Mean Relative Error | $MRE = \dfrac{1}{N}\sum_{i=1}^{N}\dfrac{S_i - O_i}{O_i}$ | $(-\infty, \infty)$ | 0 |
| Relative Error in Volume | $REV = \dfrac{\sum S_i - \sum O_i}{\sum O_i} \times 100$ | $(-\infty, \infty)$ | 0 |
| Root Mean Square Error | $RMSE = \sqrt{\dfrac{\sum_{i=1}^{N}(O_i - S_i)^2}{N}}$ | $(0, \infty)$ | 0 |
| Bias | $Bias = \dfrac{1}{N}\sum_{i=1}^{N} O_i - S_i$ | $(0, \infty)$ | 0 |
| Correlation | $Correlation = \dfrac{\sum_{i=1}^{N}(O_i - \bar{O})(S_i - \bar{S})}{\sqrt{\sum_{i=1}^{N}(O_i - \bar{O})^2 \sum_{i=1}^{N}(S_i - \bar{S})^2}}$ | $(-1, 1)$ | 1 |

10  page 18, table 3, additional figure for the statistics is more efficient for the readers

➢ *We added more figures to show the results with more graphical details. The comparison of the observed and forecast rainfall is shown for each individual station for the three events in Imjin basin.*

[Figure]

Figure 6: comparison of the accumulated observed and forecast precipitation for the events 2002, 2007 and 2011

* page 1, figure 1, location of rain gauges is necessary

*The new figure is added to indicate location of the rain gauges for different events.*

* *page 28, figure 6, legend should be used same scale*
  ➢ *The legend of the observed and forecast data is in the same scale, if we use the same legend for all the figures due to differences between maximum values, variation of the observed and forecast values cannot be seen.*

* page 33, figure 11, Y-axis subtitle is misspelled.
  ➢ *We modified the Y-axis subtitle. It is changed to forecast rainfall.*

* page 12, figure 12, unit is missed in Y-axis
  ➢ *We modified the Y-axis unit.*

---

## Author Comment (AC2) · 29 May 2018

The authors appreciate the careful review and constructive suggestions and thank the reviewers for the effort and time put into the review of the manuscript. Each comment has been carefully considered and responded in italic format. It is our belief that the manuscript is substantially improved after making the suggested edits.

Since our response file includes figures and tables, we uploaded the Author's responses to referee#2 as a .pdf file as a supplement to the comments.

Thank you for your review of our paper. We have answered each of your points in italic

format.

Please also note the supplement to this comment:
https://www.nat-hazards-earth-syst-sci-discuss.net/nhess-2017-447/nhess-2017-447-AC2-supplement.pdf

**Supplement:**

10

**Response to Referee 2:**

*Thank you for your review of our paper. We have answered each of your points below in italic format.*

15 The paper describes evaluation of WRF-downscaled driving data used in a hydrological model in a catchment on the border between North and South Korea. The paper sets out to analyse the optimal setting of temporal and spatial resolution of the precipitation modelling to produce a good hydro-meteorological forecast for the area. However, the analysis is only done over three case studies and the results do not support the conclusions. The sample size is simply too small. Further, the paper is not well organized and I am often confused by the

20 description of the methods and results; and there is not enough relevant references. I therefore do not recommend the paper to be published in its current form.

➢ *The present study has two objectives. The first objective of this study is to find proper hydrological model to couple with a meteorological model. We used the real-time forecast data of the WRF model to find the proper*

25 *hydrological model. The second objective of this study is to evaluate the effects of lead time, spatial and temporal resolution variation of the WRF model data on the performance of coupled hydro-meteorological models. A variety of tests are conducted to support our objectives in this study.*

Specific comments.

30 1. As stated already the paper would be more interesting if the study were done over a longer time period to support a robust statistical analysis of the results. It is OK to highlight certain features through case studies, but you cannot build your results on only three cases.

➢ *The authors partially agreed with reviewer's comment. We totally used nine events in this study, six events*

35 *for the calibration and verification of the hydrological model and three events used for the real-time flood forecasting by coupling SURR-WRF model. The number of events used in hydro-meteorological studies strongly depends on the purpose of the study. In this study we have two objectives. The first objective is*

*grounded on the idea to find the proper type of the hydrological model to couple with a meteorological model for a real-time flood forecasting. The results of this part led to choose the semi-distributed hydrological model. The results of the comparing the point scale and catchment scale of the precipitation analysis supports our first objective to find the proper hydrological model. In order to follow our first objective the number of events provided the required information to get a judgment for choosing the proper hydrological model.*

➤ *The second objective is to evaluate the effects of lead time, spatial and temporal resolution variation of the WRF model data on the performance of coupled hydro-meteorological models. For our second objective the real-time forecast accuracy variation assessment is done by considering the effects of the lead time forecast, spatial and temporal resolution of the meteorological model. We used three flood events which are the most important floods for the study area. We considered these events to have a reasonable chance of seeing such that effects and to provide the required sample size for comparative analysis for this research. The precipitation analysis and discharge evaluation for different spatial resolutions showed that by decreasing the spatial resolution the accuracy of the forecast decreased (following figures). Therefore it can be found that by increasing the number of events this trend will not change.*

[Figure]

➤ *Also the temporal resolution variation showed that the forecast accuracy in precipitation and discharge analysis did not change (following figures). Therefore increasing the number of events will not change this trend of accuracy variation.*

[Figure]

[Figure]

[Figure]

➢ *Also for the forecast lead-time assessment, the forecast skill decreases with the increase in lead time (following figure) and increasing the number of events will follow this trend too.*

[Figure]

5  ➢ *These events are considered to support the second objective and proposed analysis of our research. The number of three events is considered for comparison of the results and this number provides the specific aims of the analysis.*

2. The text is often too long and some details described to carefully. I also sometimes struggle to understand what
10  the authors mean, so the paper would need a thorough language revision.

➢ *The manuscript is edited previously by English Language Editing Service. However it is necessary to rearrange the sentences by breaking the long sentences into shorter ones and simplifying the sentences to reduce the confusion. The modified sentences are shown in red colour in the manuscript text. After this*
15  *review process we will again send the manuscript to the English editing.*

3. There is a lack of relevant references to support the statements made in the introduction, and the references are often too old.

➢ *Some of the old references are deleted and instead newer references related to this study are added to the manuscript. The modified sentences are shown in red in the manuscript text.*

5